# Almost Optimal Algorithms for Linear Stochastic Bandits with Heavy-Tailed Payoffs

**Han Shao**[*]  **Xiaotian Yu**[*]  **Irwin King**  **Michael R. Lyu**
Department of Computer Science and Engineering
The Chinese University of Hong Kong
{hshao,xtyu,king,lyu}@cse.cuhk.edu.hk

## Abstract

In linear stochastic bandits, it is commonly assumed that payoffs are with sub-Gaussian noises. In this paper, under a weaker assumption on noises, we study the problem of linear stochastic bandits with heavy-tailed payoffs (LinBET), where the distributions have finite moments of order $1 + \epsilon$, for some $\epsilon \in (0, 1]$. We rigorously analyze the regret lower bound of LinBET as $\Omega(T^{\frac{1}{1+\epsilon}})$, implying that finite moments of order 2 (i.e., finite variances) yield the bound of $\Omega(\sqrt{T})$, with $T$ being the total number of rounds to play bandits. The provided lower bound also indicates that the state-of-the-art algorithms for LinBET are far from optimal. By adopting median of means with a well-designed allocation of decisions and truncation based on historical information, we develop two novel bandit algorithms, where the regret upper bounds match the lower bound up to polylogarithmic factors. To the best of our knowledge, we are the first to solve LinBET optimally in the sense of the polynomial order on $T$. Our proposed algorithms are evaluated based on synthetic datasets, and outperform the state-of-the-art results.

## 1   Introduction

The decision-making model named Multi-Armed Bandits (MAB), where at each time step an algorithm chooses an arm among a given set of arms and then receives a stochastic payoff with respect to the chosen arm, elegantly characterizes the tradeoff between exploration and exploitation in sequential learning. The algorithm usually aims at maximizing cumulative payoffs over a sequence of rounds. A natural and important variant of MAB is linear stochastic bandits with the expected payoff of each arm satisfying a linear mapping from the arm information to a real number. The model of linear stochastic bandits enjoys some good theoretical properties, e.g., there exists a closed-form solution of the linear mapping at each time step in light of ridge regression. Many practical applications take advantage of MAB and its variants to control decision performance, e.g., online personalized recommendations (Li et al., 2010) and resource allocations (Lattimore et al., 2014).

In most previous studies of MAB and linear stochastic bandits, a common assumption is that noises in observed payoffs are sub-Gaussian conditional on historical information (Abbasi-Yadkori et al., 2011; Bubeck et al., 2012), which encompasses cases of all bounded payoffs and many unbounded payoffs, e.g., payoffs of an arm following a Gaussian distribution. However, there do exist practical scenarios of non-sub-Gaussian noises in observed payoffs for sequential decisions, such as high-probability extreme returns in investments for financial markets (Cont and Bouchaud, 2000) and fluctuations of neural oscillations (Roberts et al., 2015), which are called heavy-tailed noises. Thus, it is significant to completely study theoretical behaviours of sequential decisions in the case of heavy-tailed noises.

---

[*]The first two authors contributed equally.

Many practical distributions, e.g., Pareto distributions and Weibull distributions, are heavy-tailed, which perform high tail probabilities compared with exponential family distributions. We consider a general characterization of heavy-tailed payoffs in bandits, where the distributions have finite moments of order $1 + \epsilon$, where $\epsilon \in (0, 1]$. When $\epsilon = 1$, stochastic payoffs are generated from distributions with finite variances. When $0 < \epsilon < 1$, stochastic payoffs are generated from distributions with infinite variances (Shao and Nikias, 1993). Note that, different from sub-Gaussian noises in the traditional bandit setting, noises from heavy-tailed distributions do not enjoy exponentially decaying tails, and thus make it more difficult to learn a parameter of an arm.

The regret of MAB with heavy-tailed payoffs has been well addressed by Bubeck et al. (2013), where stochastic payoffs have bounds on raw or central moments of order $1 + \epsilon$. For MAB with finite variances (i.e., $\epsilon = 1$), the regret of truncation algorithms or median of means recovers the optimal regret for MAB under the sub-Gaussian assumption. Recently, Medina and Yang (2016) investigated theoretical guarantees for the problem of linear stochastic bandits with heavy-tailed payoffs (LinBET). It is surprising to find that, for $\epsilon = 1$, the regret of bandit algorithms by Medina and Yang (2016) to solve LinBET is $\widetilde{O}(T^{\frac{3}{4}})$ [2], which is far away from the regret of the state-of-the-art algorithms (i.e., $\widetilde{O}(\sqrt{T})$) in linear stochastic bandits under the sub-Gaussian assumption (Dani et al., 2008a; Abbasi-Yadkori et al., 2011). Thus, the most interesting and non-trivial question is

*Is it possible to recover the regret of $\widetilde{O}(\sqrt{T})$ when $\epsilon = 1$ for LinBET?*

In this paper, we answer this question affirmatively. Specifically, we investigate the problem of LinBET characterized by finite $(1 + \epsilon)$-th moments, where $\epsilon \in (0, 1]$. The problem of LinBET raises several interesting challenges. The first challenge is the lower bound of the problem, which remains unknown. The technical issues come from the construction of an elegant setting for LinBET, and the derivation of a lower bound with respect to $\epsilon$. The second challenge is how to develop a robust estimator for the parameter in LinBET, because heavy-tailed noises greatly affect errors of the conventional least-squares estimator. It is worth mentioning that Medina and Yang (2016) has tried to tackle this challenge, but their estimators do not make full use of the contextual information of chosen arms to eliminate the effect from heavy-tailed noises, which eventually leads to large regrets. The third challenge is how to successfully adopt median of means and truncation to solve LinBET with regret upper bounds matching the lower bound as closely as possible.

**Our Results.** First of all, we rigorously analyze the lower bound on the problem of LinBET, which enjoys a polynomial order on $T$ as $\Omega(T^{\frac{1}{1+\epsilon}})$. The lower bound provides two essential hints: one is that finite variances in LinBET yield a bound of $\Omega(\sqrt{T})$, and the other is that algorithms by Medina and Yang (2016) are sub-optimal. Then, we develop two novel bandit algorithms to solve LinBET based on the basic techniques of median of means and truncation. Both the algorithms adopt the optimism in the face of uncertainty principle, which is common in bandit problems (Abbasi-Yadkori et al., 2011; Munos et al., 2014). The regret upper bounds of the proposed two algorithms, which are $\widetilde{O}(T^{\frac{1}{1+\epsilon}})$, match the lower bound up to polylogarithmic factors. To the best of our knowledge, we are the first to solve LinBET almost optimally. We conduct experiments based on synthetic datasets, which are generated by Student's $t$-distribution and Pareto distribution, to demonstrate the effectiveness of our algorithms. Experimental results show that our algorithms outperform the state-of-the-art results. The contributions of this paper are summarized as follows:

- We provide the lower bound for the problem of LinBET characterized by finite $(1 + \epsilon)$-th moments, where $\epsilon \in (0, 1]$. In the analysis, we construct an elegant setting of LinBET, which results in a regret bound of $\Omega(T^{\frac{1}{1+\epsilon}})$ in expectation for any bandit algorithm.
- We develop two novel bandit algorithms, which are named as MENU and TOFU (with details shown in Section 4). The MENU algorithm adopts median of means with a well-designed allocation of decisions and the TOFU algorithm adopts truncation via historical information. Both algorithms achieve the regret $\widetilde{O}(T^{\frac{1}{1+\epsilon}})$ with high probability.
- We conduct experiments based on synthetic datasets to demonstrate the effectiveness of our proposed algorithms. By comparing our algorithms with the state-of-the-art results, we show improvements on cumulative payoffs for MENU and TOFU, which are strictly consistent with theoretical guarantees in this paper.

## 2 Preliminaries and Related Work

In this section, we first present preliminaries, i.e., notations and learning setting of LinBET. Then, we give a detailed discussion on the line of research for bandits with heavy-tailed payoffs.

### 2.1 Notations

For a positive integer $K$, $[K] \triangleq \{1, 2, \cdots, K\}$. Let the $\ell$-norm of a vector $x \in \mathbb{R}^d$ be $\|x\|_\ell \triangleq (x_1^\ell + \cdots + x_d^\ell)^{\frac{1}{\ell}}$, where $\ell \geq 1$ and $x_i$ is the $i$-th element of $x$ with $i \in [d]$. For $r \in \mathbb{R}$, its absolute value is $|r|$, its ceiling integer is $\lceil r \rceil$, and its floor integer is $\lfloor r \rfloor$. The inner product of two vectors $x, y$ is denoted by $x^\top y = \langle x, y \rangle$. Given a positive definite matrix $A \in \mathbb{R}^{d \times d}$, the weighted Euclidean norm of a vector $x \in \mathbb{R}^d$ is $\|x\|_A = \sqrt{x^\top A x}$. $\mathbb{B}(x, r)$ denotes a Euclidean ball centered at $x$ with radius $r \in \mathbb{R}_+$, where $\mathbb{R}_+$ is the set of positive numbers. Let $e$ be Euler's number, and $I_d \in \mathbb{R}^{d \times d}$ an identity matrix. Let $\mathbb{1}_{\{.\}}$ be an indicator function, and $\mathbb{E}[X]$ the expectation of $X$.

### 2.2 Learning Setting

For a bandit algorithm $\mathcal{A}$, we consider sequential decisions with the goal to maximize cumulative payoffs, where the total number of rounds for playing bandits is $T$. For each round $t = 1, \cdots, T$, the bandit algorithm $\mathcal{A}$ is given a decision set $D_t \subseteq \mathbb{R}^d$ such that $\|x\|_2 \leq D$ for any $x \in D_t$. $\mathcal{A}$ has to choose an arm $x_t \in D_t$ and then observes a stochastic payoff $y_t(x_t)$. For notation simplicity, we also write $y_t = y_t(x_t)$. The expectation of the observed payoff for the chosen arm satisfies a linear mapping from the arm to a real number as $y_t(x_t) \triangleq \langle x_t, \theta_* \rangle + \eta_t$, where $\theta_*$ is an underlying parameter with $\|\theta_*\|_2 \leq S$ and $\eta_t$ is a random noise. Without loss of generality, we assume $\mathbb{E}[\eta_t | \mathcal{F}_{t-1}] = 0$, where $\mathcal{F}_{t-1} \triangleq \{x_1, \cdots, x_t\} \cup \{\eta_1, \cdots, \eta_{t-1}\}$ is a $\sigma$-filtration and $\mathcal{F}_0 = \emptyset$. Clearly, we have $\mathbb{E}[y_t(x_t) | \mathcal{F}_{t-1}] = \langle x_t, \theta_* \rangle$. For an algorithm $\mathcal{A}$, to maximize cumulative payoffs is equivalent to minimizing the regret as

$$R(\mathcal{A}, T) \triangleq \left( \sum_{t=1}^T \langle x_t^*, \theta_* \rangle \right) - \left( \sum_{t=1}^T \langle x_t, \theta_* \rangle \right) = \sum_{t=1}^T \langle x_t^* - x_t, \theta_* \rangle, \tag{1}$$

where $x_t^*$ denotes the optimal decision at time $t$ for $\theta_*$, i.e., $x_t^* \in \arg\max_{x \in D_t} \langle x, \theta_* \rangle$. In this paper, we will provide high-probability upper bound of $R(\mathcal{A}, T)$ with respect to $\mathcal{A}$, and provide the lower bound for LinBET in expectation for any algorithm. The problem of LinBET is defined as below.

**Definition 1** (LinBET). *Given a decision set $D_t$ for time step $t = 1, \cdots, T$, an algorithm $\mathcal{A}$, of which the goal is to maximize cumulative payoffs over $T$ rounds, chooses an arm $x_t \in D_t$. With $\mathcal{F}_{t-1}$, the observed stochastic payoff $y_t(x_t)$ is conditionally heavy-tailed, i.e., $\mathbb{E}\left[|y_t|^{1+\epsilon}|\mathcal{F}_{t-1}\right] \leq b$ or $\mathbb{E}\left[|y_t - \langle x_t, \theta_* \rangle|^{1+\epsilon}|\mathcal{F}_{t-1}\right] \leq c$, where $\epsilon \in (0, 1]$, and $b, c \in (0, +\infty)$.*

### 2.3 Related Work

The model of MAB dates back to 1952 with the original work by Robbins et al. (1952), and its inherent characteristic is the trade-off between exploration and exploitation. The asymptotic lower bound of MAB was developed by Lai and Robbins (1985), which is logarithmic with respect to the total number of rounds. An important technique called upper confidence bound was developed to achieve the lower bound (Lai and Robbins, 1985; Agrawal, 1995). Other related techniques to solve the problem of sequential decisions include Thompson sampling (Thompson, 1933; Chapelle and Li, 2011; Agrawal and Goyal, 2012) and Gittins index (Gittins et al., 2011).

The problem of MAB with heavy-tailed payoffs characterized by finite $(1 + \epsilon)$-th moments has been well investigated (Bubeck et al., 2013; Vakili et al., 2013; Yu et al., 2018). Bubeck et al. (2013) pointed out that finite variances in MAB are sufficient to achieve regret bounds of the same order as the optimal regret for MAB under the sub-Gaussian assumption, and the order of $T$ in regret bounds increases when $\epsilon$ decreases. The lower bound of MAB with heavy-tailed payoffs has been analyzed (Bubeck et al., 2013), and robust algorithms by Bubeck et al. (2013) are optimal. Theoretical guarantees by Bubeck et al. (2013); Vakili et al. (2013) are for the setting of finite arms. In Vakili et al. (2013), primary theoretical results were presented for the case of $\epsilon > 1$. We notice that the case of $\epsilon > 1$ is not interesting, because it reduces to the case of finite variances in MAB.

For the problem of linear stochastic bandits, which is also named linear reinforcement learning by Auer (2002), the lower bound is $\Omega(d\sqrt{T})$ when contextual information of arms is from a $d$-dimensional space (Dani et al., 2008b). Bandit algorithms matching the lower bound up to poly-logarithmic factors have been well developed (Auer, 2002; Dani et al., 2008a; Abbasi-Yadkori et al., 2011; Chu et al., 2011). Notice that all these studies assume that stochastic payoffs contain sub-Gaussian noises. More variants of MAB can be discussed by Bubeck et al. (2012).

It is surprising to find that the lower bound of LinBET remains unknown. In Medina and Yang (2016), bandit algorithms based on truncation and median of means were presented. When $\epsilon$ is finite for LinBET, the algorithms by Medina and Yang (2016) cannot recover the bound of $\widetilde{O}(\sqrt{T})$ which is the regret of the state-of-the-art algorithms in linear stochastic bandits under the sub-Gaussian assumption. Medina and Yang (2016) conjectured that it is possible to recover $\widetilde{O}(\sqrt{T})$ with $\epsilon$ being a finite number for LinBET. Thus, it is urgent to conduct a thorough analysis of the conjecture in consideration of the importance of heavy-tailed noises in real scenarios. Solving the conjecture generalizes the practical applications of bandit models. Practical motivating examples for bandits with heavy-tailed payoffs include delays in end-to-end network routing (Liebeherr et al., 2012) and sequential investments in financial markets (Cont and Bouchaud, 2000).

Recently, the assumption in stochastic payoffs of MAB was relaxed from sub-Gaussian noises to bounded kurtosis (Lattimore, 2017), which can be viewed as an extension of Bubeck et al. (2013). The interesting point of Lattimore (2017) is the scale free algorithm, which might be practical in applications. Besides, Carpentier and Valko (2014) investigated extreme bandits, where stochastic payoffs of MAB follow Fréchet distributions. The setting of extreme bandits fits for the real scenario of anomaly detection without contextual information. The order of regret in extreme bandits is characterized by distributional parameters, which is similar to the results by Bubeck et al. (2013).

It is worth mentioning that, for linear regression with heavy-tailed noises, several interesting studies have been conducted. Hsu and Sabato (2016) proposed a generalized method in light of median of means for loss minimization with heavy-tailed noises. Heavy-tailed noises in Hsu and Sabato (2016) might come from contextual information, which is more complicated than the setting of stochastic payoffs in this paper. Therefore, linear regression with heavy-tailed noises usually requires a finite fourth moment. In Audibert et al. (2011), the basic technique of truncation was adopted to solve robust linear regression in the absence of exponential moment condition. The related studies in this line of research are not directly applicable for the problem of LinBET.

## 3 Lower Bound

In this section, we provide the lower bound for LinBET. We consider heavy-tailed payoffs with finite $(1 + \epsilon)$-th raw moments in the analysis. In particular, we construct the following setting. Assume $d \geq 2$ is even (when $d$ is odd, similar results can be easily derived by considering the first $d - 1$ dimensions). For $D_t \subseteq \mathbb{R}^d$ with $t \in [T]$, we fix the decision set as $D_1 = \cdots = D_T = D_{(d)}$. Then, the fixed decision set is constructed as $D_{(d)} \triangleq \{(x_1, \cdots, x_d) \in \mathbb{R}_+^d : x_1 + x_2 = \cdots = x_{d-1} + x_d = 1\}$, which is a subset of intersection of the cube $[0,1]^d$ and the hyperplane $x_1 + \cdots + x_d = d/2$. We define a set $S_d \triangleq \{(\theta_1, \cdots, \theta_d) : \forall i \in [d/2], (\theta_{2i-1}, \theta_{2i}) \in \{(2\Delta, \Delta), (\Delta, 2\Delta)\}\}$ with $\Delta \in (0, 1/d]$. The payoff functions take values in $\{0, (1/\Delta)^{\frac{1}{\epsilon}}\}$ such that, for every $x \in D_{(d)}$, the expected payoff is $\theta_*^\top x$. To be more specific, we have the payoff function of $x$ as

$$y(x) = \begin{cases} \left(\frac{1}{\Delta}\right)^{\frac{1}{\epsilon}} & \text{with a probability of } \Delta^{\frac{1}{\epsilon}} \theta_*^\top x, \\ 0 & \text{with a probability of } 1 - \Delta^{\frac{1}{\epsilon}} \theta_*^\top x. \end{cases} \quad (2)$$

We have the theorem for the lower bound of LinBET as below.

**Theorem 1** (Lower Bound of LinBET). *If $\theta_*$ is chosen uniformly at random from $S_d$, and the payoff for each $x \in D_{(d)}$ is in $\{0, (1/\Delta)^{\frac{1}{\epsilon}}\}$ with mean $\theta_*^\top x$, then for any algorithm $\mathcal{A}$ and every $T \geq (d/12)^{\frac{\epsilon}{1+\epsilon}}$, we have*

$$\mathbb{E}[R(\mathcal{A}, T)] \geq \frac{d}{192} T^{\frac{1}{1+\epsilon}}. \quad (3)$$

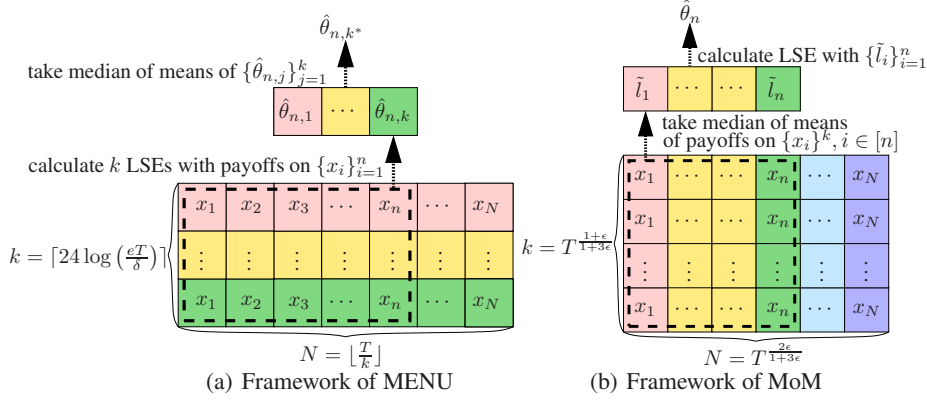

(a) Framework of MENU         (b) Framework of MoM

Figure 1: Framework comparison between our MENU and MoM by Medina and Yang (2016).

In the proof of Theorem 1, we first prove the lower bound when $d = 2$, and then generalize the argument to any $d > 2$. We notice that the parameter in the original $d$-dimensional space is rearranged to $d/2$ tuples, each of which is a 2-dimensional vector as $(\theta_{2i-1}, \theta_{2i}) \in \{(2\Delta, \Delta), (\Delta, 2\Delta)\}$ with $i \in [d/2]$. If the $i$-th tuple of the parameter is selected as $(2\Delta, \Delta)$, then the $i$-th tuple of the optimal arm is $(x_{*,2i-1}, x_{*,2i}) = (1, 0)$. In this case, if we define the $i$-th tuple of the chosen arm as $(x_{t,2i-1}, x_{t,2i})$, the instantaneous regret is $\Delta(1 - x_{t,2i-1})$. Then, the regret can be represented as an integration of $\Delta(1 - x_{t,2i-1})$ over $D_{(d)}$. Finally, with common inequalities in information theory, we obtain the regret lower bound by setting $\Delta = T^{-\frac{\epsilon}{1+\epsilon}}/12$.

We notice that martingale differences to prove the lower bound for linear stochastic bandits in (Dani et al., 2008a) are not directly feasible for the proof of lower bound in LinBET, because under our construction of heavy-tailed payoffs (i.e., Eq. (4)), the information of $\epsilon$ is excluded. Besides, our proof is partially inspired by Bubeck (2010). We show the detailed proof of Theorem 1 in Appendix A.

**Remark 1.** The above lower bound provides two essential hints: one is that finite variances in LinBET yield a bound of $\Omega(\sqrt{T})$, and the other is that algorithms proposed by Medina and Yang (2016) are far from optimal. The result in Theorem 1 strongly indicates that it is possible to design bandit algorithms recovering $\widetilde{O}(\sqrt{T})$ with finite variances.

## 4 Algorithms and Upper Bounds

In this section, we develop two novel bandit algorithms to solve LinBET, which turns out to be almost optimal. We rigorously prove regret upper bounds for the proposed algorithms. In particular, our core idea is based on the optimism in the face of uncertainty principle (OFU). The first algorithm is median of means under OFU (MENU) shown in Algorithm 1, and the second algorithm is truncation under OFU (TOFU) shown in Algorithm 2. For comparisons, we directly name the bandit algorithm based on median of means in Medina and Yang (2016) as MoM, and name the bandit algorithm based on confidence region with truncation in Medina and Yang (2016) as CRT.

Both algorithms in this paper adopt the tool of ridge regression. At time step $t$, let $\hat{\theta}_t$ be the $\ell^2$-regularized least-squares estimate (LSE) of $\theta_*$ as $\hat{\theta}_t = V_t^{-1} X_t^\top Y_t$, where $X_t \in \mathbb{R}^{t \times d}$ is a matrix of which rows are $x_1^\top, \cdots, x_t^\top$, $V_t \triangleq X_t^\top X_t + \lambda I_d$, $Y_t \triangleq (y_1, \cdots, y_t)$ is a vector of the historical observed payoffs until time $t$ and $\lambda > 0$ is a regularization parameter.

### 4.1 MENU and Regret

**Description of MENU.** To conduct median of means in LinBET, it is common to allocate $T$ pulls of bandits among $N \leq T$ epochs, and for each epoch the same arm is played multiple times to obtain an estimate of $\theta_*$. We find that there exist different ways to construct the epochs. We design the framework of MENU in Figure 1(a), and show the framework of MoM designed

---

**Algorithm 1** Median of means under OFU (MENU)

1: **input** $d, c, \epsilon, \delta, \lambda, S, T, \{D_n\}_{n=1}^N$
2: **initialization:** $k = \lceil 24 \log\left(\frac{eT}{\delta}\right)\rceil$, $N = \lfloor\frac{T}{k}\rfloor$, $V_0 = \lambda I_d$, $C_0 = \mathbb{B}(\mathbf{0}, S)$
3: **for** $n = 1, 2, \cdots, N$ **do**
4:      $(x_n, \tilde{\theta}_n) = \arg\max_{(x,\theta)\in D_n \times C_{n-1}} \langle x, \theta\rangle$           ▷ *to select an arm*
5:      Play $x_n$ with $k$ times and observe payoffs $y_{n,1}, y_{n,2}, \cdots, y_{n,k}$
6:      $V_n = V_{n-1} + x_n x_n^\top$
7:      For $j \in [k]$, $\hat{\theta}_{n,j} = V_n^{-1}\sum_{i=1}^n y_{i,j} x_i$           ▷ *to calculate LSE for the j-th group*
8:      For $j \in [k]$, let $r_j$ be the median of $\{\|\hat{\theta}_{n,j} - \hat{\theta}_{n,s}\|_{V_n} : s \in [k]\backslash j\}$
9:      $k^* = \arg\min_{j\in[k]} r_j$           ▷ *to take median of means of estimates*
10:      $\beta_n = 3\left((9dc)^{\frac{1}{1+\epsilon}} n^{\frac{1-\epsilon}{2(1+\epsilon)}} + \lambda^{\frac{1}{2}} S\right)$
11:      $C_n = \{\theta : \|\theta - \hat{\theta}_{n,k^*}\|_{V_n} \le \beta_n\}$           ▷ *to update the confidence region*
12: **end for**

---

by Medina and Yang (2016) in Figure 1(b). For MENU and MoM, we have the following three differences. First, for each epoch $n = 1, \cdots, N$, MENU plays the same arm $x_n$ by $O(\log(T))$ times, while MoM plays the same arm by $O(T^{\frac{1+\epsilon}{1+3\epsilon}})$ times. Second, at epoch $n$ with historical payoffs, MENU conducts LSEs by $O(\log(T))$ times, each of which is based on $\{x_i\}_{i=1}^n$, while MoM conducts LSE by one time based on intermediate payoffs calculated via median of means of observed payoffs. Third, MENU adopts median of means of LSEs, while MoM adopts median of means of the observed payoffs. Intuitively, the execution of multiple LSEs will lead to the improved regret of MENU. With a better trade-off between $k$ and $N$ in Figure 1(a), we derive an improved upper bound of regret in Theorem 2.

In light of Figure 1(a), we develop algorithmic procedures in Algorithm 1 for MENU. We notice that, in order to guarantee the median of means of LSEs not far away from the true underlying parameter with high probability, we construct the confidence interval in Line 10 of Algorithm 1. Now we have the following theorem for the regret upper bound of MENU.

**Theorem 2** (Regret Analysis for the MENU Algorithm). *Assume that for all $t$ and $x_t \in D_t$ with $\|x_t\|_2 \le D$, $\|\theta_*\|_2 \le S$, $|x_t^\top \theta_*| \le L$ and $\mathbb{E}[|\eta_t|^{1+\epsilon}|\mathcal{F}_{t-1}] \le c$. Then, with probability at least $1 - \delta$, for every $T \ge 256 + 24 \log(e/\delta)$, the regret of the MENU algorithm satisfies*

$$R(\text{MENU}, T) \le 6\left((9dc)^{\frac{1}{1+\epsilon}} + \lambda^{\frac{1}{2}} S + L\right) T^{\frac{1}{1+\epsilon}} \left(24 \log\left(\frac{eT}{\delta}\right) + 1\right)^{\frac{\epsilon}{1+\epsilon}} \sqrt{2d \log\left(1 + \frac{TD^2}{\lambda d}\right)}.$$

The technical challenges in MENU (i.e., Algorithm 1) and its proofs are discussed as follows. Based on the common techniques in linear stochastic bandits (Abbasi-Yadkori et al., 2011), to guarantee the instantaneous regret in LinBET, we need to guarantee $\|\theta_* - \hat{\theta}_{n,k^*}\|_{V_n} \le \beta_n$ with high probability. We attack this issue by guaranteeing $\|\theta_* - \hat{\theta}_{n,j}\|_{V_n} \le \beta_n/3$ with a probability of $3/4$, which could reduce to a problem of bounding a weighted sum of historical noises. Interestingly, by conducting singular value decomposition on $X_n$ (of which rows are $x_1^\top, \cdots, x_n^\top$), we find that 2-norm of the weights is no greater than 1. Then the weighted sum can be bounded by a term as $O\left(n^{\frac{1-\epsilon}{2(1+\epsilon)}}\right)$. With a standard analysis in linear stochastic bandits from the instantaneous regret to the regret, we achieve the above results for MENU. We show the detailed proof of Theorem 2 in Appendix B.

**Remark 2.** For MENU, we adopt the assumption of heavy-tailed payoffs on central moments, which is required in the basic technique of median of means (Bubeck et al., 2013). Besides, there exists an implicit mild assumption in Algorithm 1 that, at each epoch $n$, the decision set must contain the selected arm $x_n$ at least $k$ times, which is practical in applications, e.g., online personalized recommendations (Li et al., 2010). The condition of $T \ge 256 + 24 \log(e/\delta)$ is required for $T \ge k$. The regret upper bound of MENU is $\widetilde{O}(T^{\frac{1}{1+\epsilon}})$, which implies that finite variances in LinBET are sufficient to achieve $\widetilde{O}(\sqrt{T})$.

### 4.2 TOFU and Regret

**Description of TOFU.** We demonstrate the algorithmic procedures of TOFU in Algorithm 2. We point out two subtle differences between our TOFU and the algorithm of CRT as follows. In TOFU,

---

**Algorithm 2** Truncation under OFU (TOFU)

---
1: **input** $d, b, \epsilon, \delta, \lambda, T, \{D_t\}_{t=1}^{T}$
2: **initialization:** $V_0 = \lambda I_d, C_0 = \mathbb{B}(\mathbf{0}, S)$
3: **for** $t = 1, 2, \cdots, T$ **do**
4:     $b_t = \left(\frac{b}{\log\left(\frac{2T}{\delta}\right)}\right)^{\frac{1}{\epsilon}} t^{\frac{1-\epsilon}{2(1+\epsilon)}}$
5:     $(x_t, \tilde{\theta}_t) = \arg\max_{(x,\theta) \in D_t \times C_{t-1}} \langle x, \theta \rangle$                    ▷ *to select an arm*
6:     Play $x_t$ and observe a payoff $y_t$
7:     $V_t = V_{t-1} + x_t x_t^{\top}$ and $X_t^{\top} = [x_1, \cdots, x_t]$
8:     $[u_1, \cdots, u_d]^{\top} = V_t^{-1/2} X_t^{\top}$
9:     **for** $i = 1, \cdots, d$ **do**
10:        $Y_i^{\dagger} = (y_1 \mathbb{1}_{u_{i,1} y_1 \leq b_t}, \cdots, y_t \mathbb{1}_{u_{i,t} y_t \leq b_t})$      ▷ *to truncate the payoffs*
11:     **end for**
12:     $\theta_t^{\dagger} = V_t^{-1/2} (u_1^{\top} Y_1^{\dagger}, \cdots, u_d^{\top} Y_d^{\dagger})$
13:     $\beta_t = 4\sqrt{d} b^{\frac{1}{1+\epsilon}} \left(\log\left(\frac{2dT}{\delta}\right)\right)^{\frac{1-\epsilon}{1+\epsilon}} t^{\frac{1-\epsilon}{2(1+\epsilon)}} + \lambda^{\frac{1}{2}} S$
14:     Update $C_t = \{\theta : \|\theta - \theta_t^{\dagger}\|_{V_t} \leq \beta_t\}$                    ▷ *to update the confidence region*
15: **end for**

---

Table 1: Statistics of synthetic datasets in experiments. For Student's $t$-distribution, $\nu$ denotes the degree of freedom, $l_p$ denotes the location, $s_p$ denotes the scale. For Pareto distribution, $\alpha$ denotes the shape and $s_m$ denotes the scale. NA denotes not available.

| dataset | $D_t$ {#arms,#dimensions} | distribution {parameters} | $\{\epsilon, b, c\}$ | mean of the optimal arm |
|---|---|---|---|---|
| S1 | {20,10} | Student's $t$-distribution $\{\nu = 3, l_p = 0, s_p = 1\}$ | {1.00, NA, 3.00} | 4.00 |
| S2 | {100,20} | Student's $t$-distribution $\{\nu = 3, l_p = 0, s_p = 1\}$ | {1.00, NA, 3.00} | 7.40 |
| S3 | {20,10} | Pareto distribution $\{\alpha = 2, s_m = \frac{x_t^{\top} \theta_*}{2}\}$ | {0.50, 7.72, NA} | 3.10 |
| S4 | {100,20} | Pareto distribution $\{\alpha = 2, s_m = \frac{x_t^{\top} \theta_*}{2}\}$ | {0.50, 54.37, NA} | 11.39 |

to obtain the accurate estimate of $\theta_*$, we need to trim all historical payoffs for each dimension individually. Besides, the truncating operations depend on the historical information of arms. By contrast, in CRT, the historical payoffs are trimmed once, which is controlled only by the number of rounds for playing bandits. Compared to CRT, our TOFU achieves a tighter confidence interval, which can be found from the setting of $\beta_t$. Now we have the following theorem for the regret upper bound of TOFU.

**Theorem 3** (Regret Analysis for the TOFU Algorithm). *Assume that for all $t$ and $x_t \in D_t$ with $\|x_t\|_2 \leq D$, $\|\theta_*\|_2 \leq S$, $|x_t^{\top}\theta_*| \leq L$ and $\mathbb{E}[|y_t|^{1+\epsilon}|\mathcal{F}_{t-1}] \leq b$. Then, with probability at least $1 - \delta$, for every $T \geq 1$, the regret of the TOFU algorithm satisfies*

$$R(TOFU, T) \leq 2T^{\frac{1}{1+\epsilon}} \left(4\sqrt{d} b^{\frac{1}{1+\epsilon}} \left(\log\left(\frac{2dT}{\delta}\right)\right)^{\frac{\epsilon}{1+\epsilon}} + \lambda^{\frac{1}{2}} S + L\right) \sqrt{2d \log\left(1 + \frac{TD^2}{\lambda d}\right)}.$$

Similarly to the proof in Theorem 2, we can achieve the above results for TOFU. Due to space limitation, we show the detailed proof of Theorem 3 in Appendix C.

**Remark 3.** For TOFU, we adopt the assumption of heavy-tailed payoffs on raw moments. It is worth pointing out that, when $\epsilon = 1$, we have regret upper bound for TOFU as $\tilde{O}(d\sqrt{T})$, which implies that we recover the same order of $d$ as that under sub-Gaussian assumption (Abbasi-Yadkori et al., 2011). A weakness in TOFU is high time complexity, because for each round TOFU needs to truncate all historical payoffs. The time complexity might be reasonably reduced by dividing $T$ into multiple epochs, each of which contains only one truncation.

# 5 Experiments

In this section, we conduct experiments based on synthetic datasets to evaluate the performance of our proposed bandit algorithms: MENU and TOFU. For comparisons, we adopt two baselines: MoM and CRT proposed by Medina and Yang (2016). We run multiple independent repetitions for each dataset in a personal computer under Windows 7 with Intel CPU@3.70GHz and 16GB memory.

## 5.1 Datasets and Setting

To show effectiveness of bandit algorithms, we will demonstrate cumulative payoffs with respect to number of rounds for playing bandits over a fixed finite-arm decision set. For verifications, we adopt four synthetic datasets (named as S1–S4) in the experiments, of which statistics are shown in Table 1. The experiments on heavy tails require $\epsilon, b$ or $\epsilon, c$ to be known, which corresponds to the assumptions of Theorem 2 or Theorem 3. According to the required information, we can apply MENU or TOFU into practical applications. We adopt Student's $t$ and Pareto distributions because they are common in practice. For Student's $t$-distributions, we easily estimate $c$, while for Pareto distributions, we easily estimate $b$. Besides, we can choose different parameters (e.g., larger values) in the distributions, and recalculate the parameters of $b$ and $c$.

For S1 and S2, which contain different numbers of arms and different dimensions for the contextual information, we adopt standard Student's $t$-distribution to generate heavy-tailed noises. For the chosen arm $x_t \in D_t$, the expected payoff is $x_t^\top \theta_*$, and the observed payoff is added a noise generated from a standard Student's $t$-distribution. We generate each dimension of contextual information for an arm, as well as the underlying parameter, from a uniform distribution over $[0, 1]$. The standard Student's $t$-distribution implies that the bound for the second central moment of S1 and S2 is 3.

For S3 and S4, we adopt Pareto distribution, where the shape parameter is set as $\alpha = 2$. We know $x_t^\top \theta_* = \alpha s_m/(\alpha - 1)$ implying $s_m = x_t^\top \theta_*/2$. Then, we set $\epsilon = 0.5$ leading to the bound of raw moment as $\mathbb{E}\left[|y_t|^{1.5}\right] = \alpha s_m^{1.5}/(\alpha - 1.5) = 4s_m^{1.5}$. We take the maximum of $4s_m^{1.5}$ among all arms as the bound of the 1.5-th raw moment. We generate arms and the parameter similar to S1 and S2.

In figures, we show the average of cumulative payoffs with time evolution over ten independent repetitions for each dataset, and show error bars of a standard variance for comparing the robustness of algorithms. For S1 and S2, we run MENU and MoM and set $T = 2 \times 10^4$. For S3 and S4, we run TOFU and CRT and set $T = 1 \times 10^4$. For all algorithms, we set $\lambda = 1.0$, and $\delta = 0.1$.

## 5.2 Results and Discussions

We show experimental results in Figure 2. From the figure, we clearly find that our proposed two algorithms outperform MoM and CRT, which is consistent with the theoretical results in Theorems 2 and 3. We also evaluate our algorithms with other synthetic datasets, as well as different $\lambda$ and $\delta$, and observe similar superiority of MENU and TOFU. Finally, for further comparison on regret, complexity and storage of four algorithms, we list the results shown in Table 2.

Table 2: Comparison on regret, complexity and storage of four algorithms.

| algorithm | MoM | MENU | CRT | TOFU |
|---|---|---|---|---|
| regret | $\widetilde{O}(T^{\frac{1+2\epsilon}{1+3\epsilon}})$ | $\widetilde{O}(T^{\frac{1}{1+\epsilon}})$ | $\widetilde{O}(T^{\frac{1}{2}+\frac{1}{2(1+\epsilon)}})$ | $\widetilde{O}(T^{\frac{1}{1+\epsilon}})$ |
| complexity | $O(T)$ | $O(T \log T)$ | $O(T)$ | $O(T^2)$ |
| storage | $O(1)$ | $O(\log T)$ | $O(1)$ | $O(T)$ |

# 6 Conclusion

We have studied the problem of LinBET, where stochastic payoffs are characterized by finite $(1 + \epsilon)$-th moments with $\epsilon \in (0, 1]$. We broke the traditional assumption of sub-Gaussian noises in payoffs of bandits, and derived theoretical guarantees based on the prior information of bounds on finite moments. We rigorously analyzed the lower bound of LinBET, and developed two novel

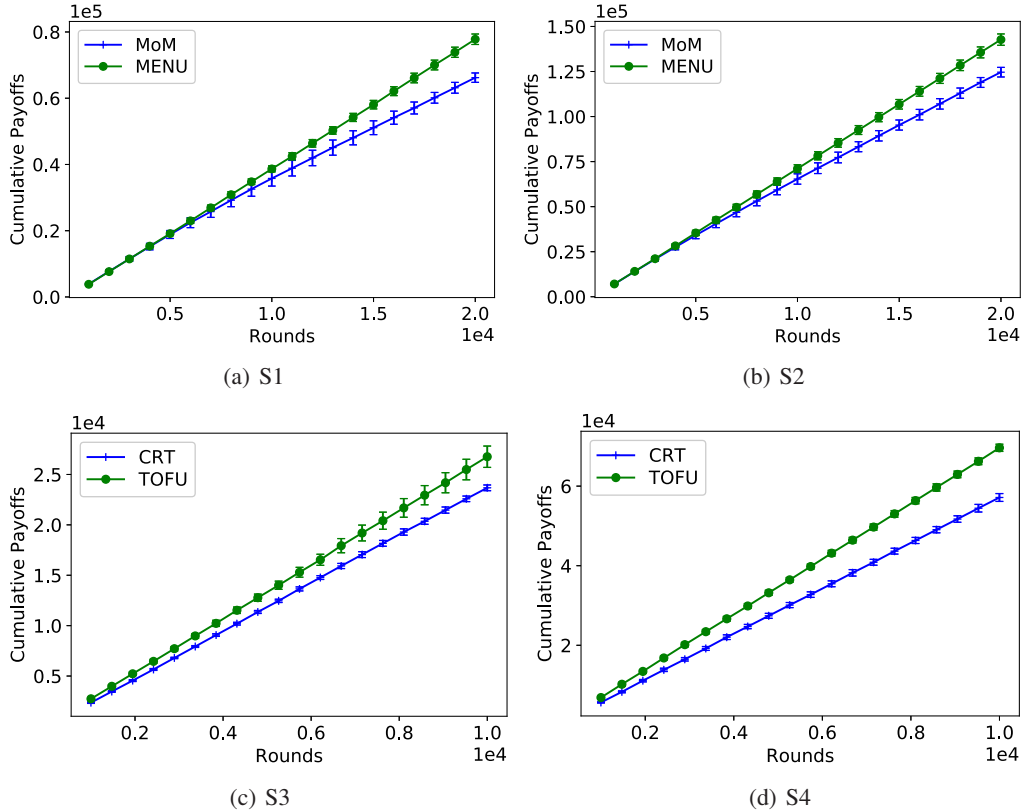

Figure 2: Comparison of cumulative payoffs for synthetic datasets S1-S4 with four algorithms.

bandit algorithms with regret upper bounds matching the lower bound up to polylogarithmic factors. Two novel algorithms were developed based on median of means and truncation. In the sense of polynomial dependence on $T$, we provided optimal algorithms for the problem of LinBET, and thus solved an open problem, which has been pointed out by Medina and Yang (2016). Finally, our proposed algorithms have been evaluated based on synthetic datasets, and outperformed the state-of-the-art results. Since both algorithms in this paper require a priori knowledge of $\epsilon$, future directions in this line of research include automatic learning of LinBET without information of distributional moments, and evaluation of our proposed algorithms in real-world scenarios.

## Acknowledgments

The work described in this paper was partially supported by the Research Grants Council of the Hong Kong Special Administrative Region, China (No. CUHK 14208815 and No. CUHK 14210717 of the General Research Fund), and Microsoft Research Asia (2018 Microsoft Research Asia Collaborative Research Award).

## Footnotes

[2] We omit polylogarithmic factors of $T$ for $\widetilde{O}(\cdot)$.

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
