[Supplementary Material]

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

## Appendix A   Proof of Theorem 1 (Lower Bound of LinBET)

We prove the lower bound for $d \geq 2$. Assume $d$ is even (when $d$ is odd, similar results can be easily derived by considering the first $d-1$ dimensions). For $D_t \subseteq \mathbb{R}^d$ with $t \in [T]$, we fix the decision set as $D_1 = \cdots = D_T = D_{(d)}$. Then, the fixed decision set is constructed as $D_{(d)} \triangleq \{(x_1, \cdots, x_d) \in \mathbb{R}^d_+ : x_1 + x_2 = \cdots = x_{d-1} + x_d = 1\}$, which is a subset of intersection of the cube $[0,1]^d$ and the hyperplane $x_1 + \cdots + x_d = d/2$. We define a set $S_d \triangleq \{(\theta_1, \cdots, \theta_d) : \forall i \in [d/2], (\theta_{2i-1}, \theta_{2i}) \in \{(2\Delta, \Delta), (\Delta, 2\Delta)\}\}$ with $\Delta \in (0, 1/d]$. The payoff functions take values in $\{0, (1/\Delta)^{\frac{1}{\epsilon}}\}$ with $\epsilon \in (0,1]$, for every $x \in D_{(d)}$, the expected payoff is $\theta_*^\top x$, where $\theta_*$ is the underlying parameter drawn from $S_d$. To be more specific, we have the payoff function of $x$ as

$$y(x) = \begin{cases} \left(\frac{1}{\Delta}\right)^{\frac{1}{\epsilon}} & \text{with a probability of } \Delta^{\frac{1}{\epsilon}} \theta_*^\top x, \\ 0 & \text{with a probability of } 1 - \Delta^{\frac{1}{\epsilon}} \theta_*^\top x. \end{cases} \tag{4}$$

In this setting, the $(1+\epsilon)$-th raw moments of payoffs are bounded by $d$ and $|\theta_*^\top x| \leq 1$. We start the proof with the 2-dimensional case in Subsection A.1. Its extension to the general case (i.e., $d > 2$) is provided in Subsection A.2. Though we set a fixed decision set in the proofs, we can easily extend the lower bound here to the setting of time-varying decision sets, as discussed by Dani et al. (2008a).

### A.1   $d = 2$ Case

Let $\mu_0 = (\Delta, \Delta)$, $\mu_1 = (2\Delta, \Delta)$ and $\mu_2 = (\Delta, 2\Delta)$. The 2-dimensional decision set is $D_{(2)} = \{(x_1, x_2) \in \mathbb{R}^2_+ : x_1 + x_2 = 1\}$. Our payoff functions take values in $\{0, (1/\Delta)^{\frac{1}{\epsilon}}\}$, and for every $x \in D_{(2)}$, the expected payoff is $\theta_*^\top x$, where $\theta_*$ is chosen uniformly at random from $\{\mu_1, \mu_2\}$. It is easy to find $\mu_j^\top x = \Delta(1 + x_j)$ which is maximized at $x_j = 1$ for $j \in \{1, 2\}$, and $\mu_0^\top x = \Delta$ for any $x \in D_{(2)}$.

**Lemma 1.** *If $\theta_*$ is chosen uniformly at random from $\{\mu_1, \mu_2\}$, and the payoff for each $x \in D_{(2)}$ is in $\{0, (1/\Delta)^{\frac{1}{\epsilon}}\}$ with mean $\theta_*^\top x$, then for every algorithm $\mathcal{A}$ and every $T \geq 1$, the regret satisfies*

$$\mathbb{E}[R(\mathcal{A}, T)] \geq \frac{1}{96} T^{\frac{1}{1+\epsilon}}. \tag{5}$$

*Proof.* We consider a deterministic algorithm $\mathcal{A}$ first. Let $q_{x,T} = T(x)/T$, where $T(x)$ denotes the number of pulls of arm $x$. $\mathbb{Q}_T$ is the empirical distribution of arms with respect to $q_{x,T}$ and $X$ is drawn from $\mathbb{Q}_T$. We let $\mathbb{P}_j$ and $\mathbb{E}_j$ denote, respectively, the probability distribution of $X$ conditional on $\theta_* = \mu_j$ and the expectation conditional on $\theta_* = \mu_j$, where $j \in \{0, 1, 2\}$. Thus, we have $\mathbb{P}_j(X \in \mathcal{E}) = \mathbb{E}_j[\sum_{x \in \mathcal{E}} T(x)]/T$ for any $\mathcal{E} \subseteq D_{(2)}$. At each time step $t$, $x_t = (x_{t,1}, x_{t,2})$ is selected. We let $y_t^* = \langle x_t^*, \theta_* \rangle$. Hence, for $j \in \{1, 2\}$, we have

$$\mathbb{E}_j \left[ \sum_{t=1}^T (y_t^* - y_t(x_t)) \right] = \sum_{t=1}^T \mathbb{E}_j [\Delta(1 - x_{t,j})] = T \int_{D_{(2)}} \Delta(1 - x_j) d\mathbb{P}_j(x)$$

$$= T\Delta \left( 1 - \int_{D_{(2)}} x_j d\mathbb{P}_j(x) \right) = T\Delta \left( 1 - \left( \int_{0 \leq x_j \leq \frac{1}{2}} x_j d\mathbb{P}_j(x) + \int_{\frac{1}{2} < x_j \leq 1} x_j d\mathbb{P}_j(x) \right) \right)$$

$$\geq T\Delta \left( 1 - \left( \frac{1}{2} \mathbb{P}_j \left( 0 \leq X_j \leq \frac{1}{2} \right) + \mathbb{P}_j \left( \frac{1}{2} < X_j \leq 1 \right) \right) \right), \tag{6}$$

which implies

$$\mathbb{E}[R(\mathcal{A}, T)] = \mathbb{E}_{\theta_*} \left[ \mathbb{E}_j \left[ \sum_{t=1}^T (y_t^* - y_t(x_t)) \right] \right]$$

$$\geq T\Delta \left( 1 - \frac{1}{2} \sum_{j=1}^2 \left( \frac{1}{2} \mathbb{P}_j \left( 0 \leq X_j \leq \frac{1}{2} \right) + \mathbb{P}_j \left( \frac{1}{2} < X_j \leq 1 \right) \right) \right). \tag{7}$$

According to Pinsker's inequality, for any $\mathcal{E} \subseteq D_{(2)}$, we have

$$\mathbb{P}_j(X \in \mathcal{E}) \leq \mathbb{P}_0(X \in \mathcal{E}) + \sqrt{\frac{1}{2}\mathrm{KL}(\mathbb{P}_0, \mathbb{P}_j)}, \tag{8}$$

where $\mathrm{KL}(\mathbb{P}_0, \mathbb{P}_j)$ denotes the Kullback-Leibler divergence (simply KL divergence). Hence,

$$\mathbb{E}[R(\mathcal{A}, T)] \geq T\Delta \left(1 - \frac{1}{2}\sum_{j=1}^{2}\left(\frac{1}{2}\mathbb{P}_0\left(0 \leq X_j \leq \frac{1}{2}\right) + \mathbb{P}_0\left(\frac{1}{2} < X_j \leq 1\right) + \frac{3}{2}\sqrt{\frac{1}{2}\mathrm{KL}(\mathbb{P}_0, \mathbb{P}_j)}\right)\right)$$

$$= T\Delta\left(\frac{1}{4} - \frac{3}{4}\sum_{j=1}^{2}\sqrt{\frac{1}{2}\mathrm{KL}(\mathbb{P}_0, \mathbb{P}_j)}\right). \tag{9}$$

Since $\mathcal{A}$ is deterministic, the sequence of received rewards $W_T \triangleq (y_1, y_2, \cdots, y_T) \in \{0, (1/\Delta)^{\frac{1}{\epsilon}}\}^T$ uniquely determines the empirical distribution $\mathbb{Q}_T$ and thus, $\mathbb{Q}_T$ conditional on $W_T$ is the same for any $\theta_*$. We let $\mathbb{P}_j^t$ be the probability distribution of $W_t = (y_1, y_2, \cdots, y_t)$ conditional on $\theta_* = \mu_j$. Based on the chain rule for KL divergence, we have

$$\mathrm{KL}(\mathbb{P}_0, \mathbb{P}_j) \leq \mathrm{KL}(\mathbb{P}_0^T, \mathbb{P}_j^T). \tag{10}$$

Further, iteratively using the chain rule for KL divergence, we have

$$\mathrm{KL}(\mathbb{P}_0^T, \mathbb{P}_j^T) = \mathrm{KL}(\mathbb{P}_0^1, \mathbb{P}_j^1) + \sum_{t=2}^{T}\int_{W_{t-1}} \mathrm{KL}\left(\mathbb{P}_0^t(\cdot|w_{t-1}), \mathbb{P}_j^t(\cdot|w_{t-1})\right)d\mathbb{P}_0^{t-1}(W_{t-1})$$

$$= \mathrm{KL}(\mathbb{P}_0^1, \mathbb{P}_j^1) + \tag{11}$$

$$\sum_{t=2}^{T}\int_{x_t \in D_{(2)}}\int_{W_{t-1}|x_{t,j}=x_j} \mathrm{KL}\left(\Delta^{\frac{1+\epsilon}{\epsilon}}, \Delta^{\frac{1+\epsilon}{\epsilon}}(1 + x_j)\right)d\mathbb{P}_0^{t-1}(W_{t-1}|x_{t,j}=x_j)d\mathbb{P}_0^{t-1}(x_{t,j}=x_j) \tag{12}$$

$$\leq 2\Delta^{\frac{1+\epsilon}{\epsilon}} + \sum_{t=2}^{T}\int_{x_t \in D_{(2)}}\int_{W_{t-1}|x_{t,j}=x_j} 2\Delta^{\frac{1+\epsilon}{\epsilon}}d\mathbb{P}_0^{t-1}(W_{t-1}|x_{t,j}=x_j)d\mathbb{P}_0^{t-1}(x_{t,j}=x_j) \tag{13}$$

$$= 2T\Delta^{\frac{1+\epsilon}{\epsilon}}, \tag{14}$$

where Eq. (13) could be derived by setting $\Delta \leq (1/2)^{\frac{\epsilon}{1+\epsilon}}$. Note that for any $p, q \in (0, 1)$, let $\mathbb{P}$ and $\mathbb{Q}$ denote the Bernoulli distribution with parameters $p$ and $q$ respectively. We denote $\mathrm{KL}(\mathbb{P}, \mathbb{Q})$ as $\mathrm{KL}(p, q)$ in Eq. (12). Therefore, we have

$$\mathbb{E}[R(\mathcal{A}, T)] \geq T\Delta\left(\frac{1}{4} - \frac{3}{2}\sqrt{T\Delta^{\frac{1+\epsilon}{\epsilon}}}\right) \geq \frac{1}{96}T^{\frac{1}{1+\epsilon}}, \tag{15}$$

where we set $\Delta = T^{-\frac{\epsilon}{1+\epsilon}}/12$.

So far we have discussed the case where $\mathcal{A}$ is a deterministic algorithm. When $\mathcal{A}$ is a randomized algorithm, the result is the same. In particular, let $\mathbb{E}_{\mathcal{A}}$ denote the expectation with respect to the randomness of $\mathcal{A}$. Then, we have

$$\mathbb{E}[R(\mathcal{A}, T)] = \mathbb{E}_{\mathcal{A}}\left[\mathbb{E}_{\theta_*}\left[\mathbb{E}_j\left[\sum_{t=1}^{T}(y_t^* - y_t(x_t))\right]\right]\right]. \tag{16}$$

If we fix the realization of the algorithm's randomization, the results of the previous steps for a deterministic algorithm apply and $\mathbb{E}_{\theta_*}\left[\mathbb{E}_i\left[\sum_{t=1}^{T}(y_t^* - y_t(x_t))\right]\right]$ could be lower bounded as before. Hence, $\mathbb{E}[R(\mathcal{A}, T)]$ is lower bounded as Eq. (15). $\square$

## A.2 General Case ($d > 2$)

Now we suppose $d > 2$ is even. If $d$ is odd, we just take the first $d - 1$ dimensions into consideration. Then we consider the contribution to the total expected regret from the choice of $(x_{2i-1}, x_{2i})$, for all $i \in [d/2]$. We call $(x_{2i-1}, x_{2i})$ the $i$-th component of $x$.

Analogously to the $d = 2$ case, we set $(\theta_{*,2i-1}, \theta_{*,2i}) \in \{\mu_1, \mu_2\}$. The decision region is $D_{(d)} = \{(x_1, \cdots, x_d) \in \mathbb{R}^d_+ : x_1 + x_2 = \cdots = x_{d-1} + x_d = 1\}$. Then, by following the proof for $d = 2$ case, we could derive the regret due to the $i$-th component of $x$ as

$$\mathbb{E}\left[R^{(i)}(\mathcal{A}, T)\right] \geq \frac{1}{96} T^{\frac{1}{1+\epsilon}}, \tag{17}$$

where $i \in [d/2]$. Summing over the $d/2$ components of Eq. (17) completes the proof for Theorem 1.

## Appendix B    Proof of Theorem 2 (Regret Analysis for the MENU Algorithm)

To prove Theorem 2, we start with proving the following two lemmas. Recall that the algorithm in the paper is based on least-squares estimate (LSE).

**Lemma 2** (Confidence Ellipsoid of LSE). *Let $\hat{\theta}_n$ denote the LSE of $\theta_*$ with the sequence of decisions $x_1, \cdots, x_n$ and observed payoffs $y_1, \cdots, y_n$. Assume that for all $\tau \in [n]$ and all $x_\tau \in D_\tau \subseteq \mathbb{R}^d$, $\mathbb{E}[|\eta_\tau|^{1+\epsilon}|\mathcal{F}_{\tau-1}] \leq c$ and $\|\theta_*\|_2 \leq S$. Then $\hat{\theta}_n$ satisfies*

$$Pr\left(\|\hat{\theta}_n - \theta_*\|_{V_n} \leq (9dc)^{\frac{1}{1+\epsilon}} n^{\frac{1-\epsilon}{2(1+\epsilon)}} + \lambda^{\frac{1}{2}} S\right) \geq \frac{3}{4}, \tag{18}$$

*where $\lambda > 0$ is a regularization parameter and $V_n = \lambda I_d + \sum_{\tau=1}^n x_\tau x_\tau^\top$.*

*Proof.* The singular value decomposition of $X_n$ is $U\Sigma_n V^\top$, where $U$ is an $n \times d$ matrix with orthonormal columns, $V$ is a $d \times d$ unitary matrix and $\Sigma_n$ is an $n \times n$ diagonal matrix with non-negative entries. We calculate $V_n = V(\Sigma_n^2 + \lambda I_d)V^\top$ and

$$V_n^{-\frac{1}{2}} X_n^\top = V\left(\Sigma_n^2 + \lambda I_d\right)^{-\frac{1}{2}} \Sigma_n U^\top. \tag{19}$$

Let $u_i^\top$ denote the $i$-th row of $V\left(\Sigma_n^2 + \lambda I_d\right)^{-\frac{1}{2}} \Sigma_n U^\top$, which leads to $\|u_i\|_2 \leq 1$. More importantly, by optimization, we have $\|u_i\|_{1+\epsilon} \leq n^{\frac{1-\epsilon}{2(1+\epsilon)}}$. By letting $Y_n = (y_1, \cdots, y_n)$, we have

$$\|\hat{\theta}_n - \theta_*\|_{V_n} = \|V_n^{-1} X_n^\top (Y_n - X_n \theta_*) - \lambda V_n^{-1} \theta_*\|_{V_n}$$

$$\leq \|V_n^{-\frac{1}{2}} X_n^\top (Y_n - X_n \theta_*)\|_2 + \lambda \|\theta_*\|_{V_n^{-1}} \leq \sqrt{\sum_{i=1}^d \left(u_i^\top (Y_n - X_n \theta_*)\right)^2} + \lambda^{\frac{1}{2}} S. \tag{20}$$

Inspired by Bubeck et al. (2013); Medina and Yang (2016), we bound the desired probability by using a union bound as

$$Pr\left(\sum_{i=1}^d \left(\sum_{\tau=1}^n u_{i,\tau} \eta_\tau\right)^2 > \gamma^2\right) \leq Pr\left(\exists i, \tau, |u_{i,\tau} \eta_\tau| > \gamma\right) + Pr\left(\sum_{i=1}^d \left(\sum_{\tau=1}^n u_{i,\tau} \eta_\tau \mathbb{1}_{|u_{i,\tau} \eta_\tau| \leq \gamma}\right)^2 > \gamma^2\right), \tag{21}$$

where $\mathbb{1}_{\{\cdot\}}$ is the indicator function. By using a union bound and Markov's inequality, the first term could be bounded as

$$Pr\left(\exists i, \tau, |u_{i,\tau} \eta_\tau| > \gamma\right) \leq \sum_{i=1}^d \sum_{\tau=1}^n Pr(|u_{i,\tau} \eta_\tau| > \gamma) \leq \frac{\sum_{i=1}^d \sum_{\tau=1}^n \mathbb{E}[|u_{i,\tau} \eta_\tau|^{1+\epsilon}]}{\gamma^{1+\epsilon}} \tag{22}$$

$$\leq \frac{\sum_{i=1}^d \sum_{\tau=1}^n |u_{i,\tau}|^{1+\epsilon} c}{\gamma^{1+\epsilon}} \leq \frac{dc n^{\frac{1-\epsilon}{2}}}{\gamma^{1+\epsilon}}.$$

Based on Markov's inequality, we bound the second term as

$$\Pr\left(\sum_{i=1}^{d}\left(\sum_{\tau=1}^{n}u_{i,\tau}\eta_\tau\mathbb{1}_{|u_{i,\tau}\eta_\tau|\leq\gamma}\right)^2>\gamma^2\right)\leq\frac{\mathbb{E}\left[\sum_{i=1}^{d}(\sum_{\tau=1}^{n}u_{i,\tau}\eta_\tau\mathbb{1}_{|u_{i,\tau}\eta_\tau|\leq\gamma})^2\right]}{\gamma^2}$$

$$=\sum_{i=1}^{d}\left(\frac{\mathbb{E}\left[\sum_{\tau=1}^{n}(u_{i,\tau}\eta_\tau)^2\mathbb{1}_{|u_{i,\tau}\eta_\tau|\leq\gamma}\right]}{\gamma^2}+2\frac{\mathbb{E}\left[\sum_{\tau'>\tau}(u_{i,\tau}\eta_\tau)\mathbb{1}_{|u_{i,\tau}\eta_\tau|\leq\gamma}(u_{i,\tau'}\eta_{\tau'})\mathbb{1}_{|u_{i,\tau'}\eta_{\tau'}|\leq\gamma}\right]}{\gamma^2}\right)$$

$$\leq\sum_{i=1}^{d}\left(\frac{\mathbb{E}\left[\sum_{\tau=1}^{n}(u_{i,\tau}\eta_\tau)^2\mathbb{1}_{|u_{i,\tau}\eta_\tau|\leq\gamma}\right]}{\gamma^2}+2\frac{\sum_{\tau'>\tau}\mathbb{E}[(u_{i,\tau}\eta_\tau)\mathbb{1}_{|u_{i,\tau}\eta_\tau|\leq\gamma}]\mathbb{E}[(u_{i,\tau'}\eta_{\tau'})\mathbb{1}_{|u_{i,\tau'}\eta_{\tau'}|\leq\gamma}|\mu_{i,\tau}\eta_\tau]}{\gamma^2}\right)$$

$$\leq\sum_{i=1}^{d}\left(\frac{\sum_{\tau=1}^{n}|u_{i,\tau}|^{1+\epsilon}c}{\gamma^{1+\epsilon}}+\left(\frac{\sum_{\tau=1}^{n}|u_{i,\tau}|^{1+\epsilon}c}{\gamma^{1+\epsilon}}\right)^2\right) \tag{23}$$

$$\leq\frac{dcn^{\frac{1-\epsilon}{2}}}{\gamma^{1+\epsilon}}+d\left(\frac{n^{\frac{1-\epsilon}{2}}c}{\gamma^{1+\epsilon}}\right)^2. \tag{24}$$

Note that Eq. (23) uses the fact that $\mathbb{E}[(u_{i,\tau}\eta_\tau)\mathbb{1}_{|u_{i,\tau}\eta_\tau|\leq\gamma}|\mathcal{F}_{\tau-1}]=-\mathbb{E}[(u_{i,\tau}\eta_\tau)\mathbb{1}_{|u_{i,\tau}\eta_\tau|>\gamma}|\mathcal{F}_{\tau-1}]$. Finally, setting $\gamma=(9dc)^{\frac{1}{1+\epsilon}}n^{\frac{1-\epsilon}{2(1+\epsilon)}}$ completes the proof. $\square$

**Lemma 3.** *Recall $\hat{\theta}_{n,j}$, $\hat{\theta}_{n,k^*}$ and $V_n$ in MENU (i.e., Algorithm 1). If there exists a $\gamma>0$ such that $\Pr\left(\|\hat{\theta}_{n,j}-\theta_*\|_{V_n}\leq\gamma\right)\geq\frac{3}{4}$ holds for all $j\in[k]$ with $k\geq1$, then with probability at least $1-e^{-\frac{k}{24}}$, $\|\hat{\theta}_{n,k^*}-\theta_*\|_{V_n}\leq3\gamma$.*

*Proof.* The proof is inspired by Hsu and Sabato (2014). We define $b_j\triangleq\mathbb{1}_{\|\hat{\theta}_{n,j}-\theta_*\|_{V_n}>\gamma}$, $p_j\triangleq\Pr(b_j=1)$ and $\mathbb{B}_{V_n}(\theta_*,\gamma)\triangleq\{\theta:\|\theta-\theta_*\|_{V_n}\leq\gamma\}$. We know that $p_j<1/4$. By Azuma-Hoeffding's inequality, we have

$$\Pr\left(\sum_{j=1}^{k}b_j\geq\frac{k}{3}\right)<\Pr\left(\sum_{j=1}^{k}b_j-p_j\geq\frac{k}{12}\right)\leq e^{-\frac{k}{24}}, \tag{25}$$

which means that more than 2/3 of $\{\hat{\theta}_{n,1},\cdots,\hat{\theta}_{n,k}\}$ are contained in $\mathbb{B}_{V_n}(\theta_*,\gamma)$ (denoting by this event $\mathcal{E}$) with probability at least $1-e^{-\frac{k}{24}}$. Note that the value $k/3$ in Eq. (25) could also be set as other values in $(k/4,k/2)$. Conditional on the event $\mathcal{E}$, by letting $r_j$ be the median of $\{\|\hat{\theta}_{n,j}-\hat{\theta}_{n,s}\|_{V_n}:s\in[k]\backslash j\}$, we have

- If $\hat{\theta}_{n,j}\in\mathbb{B}_{V_n}(\theta_*,\gamma)$, $\|\hat{\theta}_{n,j}-\hat{\theta}_{n,s}\|_{V_n}\leq2\gamma$ for all $\hat{\theta}_{n,s}\in\mathbb{B}_{V_n}(\theta_*,\gamma)$ by triangle inequality. Therefore, $r_j\leq2\gamma$.
- If $\hat{\theta}_{n,j}\notin\mathbb{B}_{V_n}(\theta_*,3\gamma)$, $\|\hat{\theta}_{n,j}-\hat{\theta}_{n,s}\|_{V_n}>2\gamma$ for all $\hat{\theta}_{n,s}\in\mathbb{B}_{V_n}(\theta_*,\gamma)$ by triangle inequality. Therefore, $r_j>2\gamma$.

Combining the above two cases completes proof. $\square$

Based on Lemmas 2 and 3, by setting $k=\lceil24\log(eT/\delta)\rceil$, we have $\|\hat{\theta}_{n,k^*}-\theta_*\|_{V_n}\leq3\left((9dc)^{\frac{1}{1+\epsilon}}n^{\frac{1-\epsilon}{2(1+\epsilon)}}+\lambda^{\frac{1}{2}}S\right)$ with probability at least $1-\delta/T$. The following part of proof is standard (Dani et al., 2008a; Abbasi-Yadkori et al., 2011). We include it for the sake of completeness. By letting $\beta_n=3\left((9dc)^{\frac{1}{1+\epsilon}}n^{\frac{1-\epsilon}{2(1+\epsilon)}}+\lambda^{\frac{1}{2}}S\right)$, we can decompose the instantaneous regret as follows:

$$r_n=\theta_*^\top x_*-\theta_*^\top x_n\leq\tilde{\theta}_n^\top x_n-\theta_*^\top x_n\leq\left(\|\tilde{\theta}_n-\hat{\theta}_{n-1,k^*}\|_{V_{n-1}}+\|\hat{\theta}_{n-1,k^*}-\theta_*\|_{V_{n-1}}\right)\|x_n\|_{V_{n-1}^{-1}}$$

$$\leq2\beta_{n-1}\|x_n\|_{V_{n-1}^{-1}}, \tag{26}$$

where we recall that $(x_n,\tilde{\theta}_n)$ is optimistic in MENU. Note that, for $n=1$, the above inequality also holds with $V_0=\lambda I_d$. On the other hand, by considering $|x_t^\top\theta_*|\leq L$, we always have

$$r_n\leq2L. \tag{27}$$

We can get that

$$r_n \leq 2 \min\{\beta_{n-1}\|x_n\|_{V_{n-1}^{-1}}, L\} \leq 2(\beta_{n-1} + L) \min\{\|x_n\|_{V_{n-1}^{-1}}, 1\}. \tag{28}$$

Following Lemma 11 of Abbasi-Yadkori et al. (2011), we know that

$$\sum_{n=1}^{N} \min\{\|x_n\|_{V_{n-1}^{-1}}^2, 1\} \leq 2 \sum_{n=1}^{N} \log(1 + \|x_n\|_{V_{n-1}^{-1}}^2) = 2 \log\left(\frac{\det(V_N)}{\det(V_0)}\right) \leq 2d \log\left(1 + \frac{ND^2}{\lambda d}\right), \tag{29}$$

where $N$ is the number of epochs in MENU. Therefore, the total regret can be upper bounded by

$$R(\text{MENU}, T) \leq k \sum_{n=1}^{N} r_n \leq k \sqrt{N \sum_{n=1}^{N} r_n^2} \leq 2kN^{\frac{1}{2}}(\beta_N + L) \sqrt{\sum_{n=1}^{N} \min\{\|x_n\|_{V_{n-1}^{-1}}^2, 1\}}$$

$$\leq 6\left((12dc)^{\frac{1}{1+\epsilon}} + \lambda^{\frac{1}{2}}S + L\right) T^{\frac{1}{1+\epsilon}} \left(24 \log\left(\frac{eT}{\delta}\right) + 1\right)^{\frac{\epsilon}{1+\epsilon}} \sqrt{2d \log\left(1 + \frac{TD^2}{\lambda d}\right)}. \tag{30}$$

The condition of $T \geq 256 + 24 \log(e/\delta)$ is required for $T \geq k$, which completes the proof.

## Appendix C    Proof of Theorem 3 (Regret Analysis for the TOFU Algorithm)

**Lemma 4** (Confidence Ellipsoid of Truncated Estimate). *With the sequence of decisions $x_1, \cdots, x_t$, the truncated payoffs $\{Y_i^\dagger\}_{i=1}^d$ and the parameter estimate $\theta_t^\dagger$ are defined in TOFU (i.e., Algorithm 2). Assume that for all $\tau \in [t]$ and all $x_\tau \in D_\tau \subseteq \mathbb{R}^d$, $\mathbb{E}[|y_\tau|^{1+\epsilon}|\mathcal{F}_{\tau-1}] \leq b$ and $\|\theta_*\|_2 \leq S$. With probability at least $1 - \delta$, we have*

$$\|\theta_t^\dagger - \theta_*\|_{V_t} \leq 4\sqrt{d}b^{\frac{1}{1+\epsilon}} \left(\log\left(\frac{2d}{\delta}\right)\right)^{\frac{\epsilon}{1+\epsilon}} t^{\frac{1-\epsilon}{2(1+\epsilon)}} + \lambda^{\frac{1}{2}}S, \tag{31}$$

*where $\lambda > 0$ is a regularization parameter and $V_t = \lambda I_d + \sum_{\tau=1}^{t} x_\tau x_\tau^\top$.*

*Proof.* Similarly to Eq. (20), we have

$$\|\theta_t^\dagger - \theta_*\|_{V_t} \leq \sqrt{\sum_{i=1}^{d} \left(u_i^\top (Y_i^\dagger - X_t \theta_*)\right)^2} + \lambda^{\frac{1}{2}}S. \tag{32}$$

We let $y_\tau^\dagger$ denote $Y_{i,\tau}^\dagger$ for notation simplicity as the following proof holds for all $i \in [d]$. Then with probability at least $1 - \delta/d$, we have

$$u_i^\top \left(Y_i^\dagger - X_t \theta_*\right) = \sum_{\tau=1}^{t} u_{i,\tau} \left(y_\tau^\dagger - \mathbb{E}[y_\tau|\mathcal{F}_{\tau-1}]\right) \tag{33}$$

$$= \sum_{\tau=1}^{t} u_{i,\tau} \left(y_\tau^\dagger - \mathbb{E}\left[y_\tau^\dagger|\mathcal{F}_{\tau-1}\right] - \mathbb{E}\left[y_\tau \mathbb{1}_{|u_{i,\tau}y_\tau|>b_t}|\mathcal{F}_{\tau-1}\right]\right)$$

$$\leq \left|\sum_{\tau=1}^{t} u_{i,\tau}(y_\tau^\dagger - \mathbb{E}[y_\tau^\dagger|\mathcal{F}_{\tau-1}])\right| + \left|\sum_{\tau=1}^{t} u_{i,\tau}\mathbb{E}[y_\tau \mathbb{1}_{|u_{i,\tau}y_\tau|>b_t}|\mathcal{F}_{\tau-1}]\right|$$

$$\leq \left|2b_t \log\left(\frac{2d}{\delta}\right) + \frac{1}{2b_t} \sum_{\tau=1}^{t} \mathbb{E}\left[u_{i,\tau}^2 \left(y_\tau^\dagger - \mathbb{E}\left[y_\tau^\dagger|\mathcal{F}_{\tau-1}\right]\right)^2 |\mathcal{F}_{\tau-1}\right]\right| + \left|\sum_{\tau=1}^{t} \mathbb{E}[u_{i,\tau}y_\tau \mathbb{1}_{|u_{i,\tau}y_\tau|>b_t}|\mathcal{F}_{\tau-1}]\right| \tag{34}$$

$$\leq 2b_t \log\left(\frac{2d}{\delta}\right) + \frac{\sum_{\tau=1}^{t} |u_{i,\tau}|^{1+\epsilon}b}{2b_t^\epsilon} + \frac{\sum_{\tau=1}^{t} |u_{i,\tau}|^{1+\epsilon}b}{b_t^\epsilon}$$

$$\leq 4b^{\frac{1}{1+\epsilon}} \left(\log\left(\frac{2d}{\delta}\right)\right)^{\frac{\epsilon}{1+\epsilon}} t^{\frac{1-\epsilon}{2(1+\epsilon)}}, \tag{35}$$

where Eq. (34) is obtained by applying Bernstein's inequality for martingales (Seldin et al., 2012) and Eq. (35) is obtained by the fact that $\|u_i\|_{1+\epsilon} \leq t^{\frac{1-\epsilon}{2(1+\epsilon)}}$ and by setting $b_t = (b/\log(2d/\delta))^{\frac{1}{1+\epsilon}} t^{\frac{1-\epsilon}{2(1+\epsilon)}}$. Combining Eq. (32) and Eq. (35) completes the proof.  □

With similar procedures to the proof of Theorem 2, we have the regret of TOFU as:

$$R(\text{TOFU}, T) \leq 2T^{\frac{1}{1+\epsilon}} \left( 4\sqrt{d}b^{\frac{1}{1+\epsilon}} \left( \log \left( \frac{2dT}{\delta} \right) \right)^{\frac{\epsilon}{1+\epsilon}} + \lambda^{\frac{1}{2}}S + L \right) \sqrt{2d \log \left( 1 + \frac{TD^2}{\lambda d} \right)}, \qquad (36)$$

which completes the proof.