[Reviews · NeurIPS 2018]

Reviewer 1



This paper studies linear stochastic bandits with heavy-tailed payoffs. Heavy-tailed means that the distributions have finite moments of order 1+epsilon for some 0

Reviewer 2



EDIT: I read your rebuttal, which is excellent and very informative, thank you. This is great work, I hope you'll make it as reproducible as possible by posting your code online afterwards :) This paper analysis the stochastic linear bandits under heavy tailed noise models. As this is not a novel problem, the authors clearly link their work with existing algorithms and results and show how they are able to improve on these existing bounds. They prove a lower bound under (1+\epsilon)-moment assumption, analysis their algorithms both theoretically and empirically. I was able to check most of the technical material in the supplementary paper and I believe it is sound and some results are even of independent interest (confidence ellipsoids for truncated observation for instance). I recommend acceptance. Comments and questions: - Could be nice to give a sort of guideline: for which problem should I use which algorithm ? I find it not very clear in the experiments why you chose either of them for the problems you defined. - Table 1: I don’t see quite well the added value of the last two columns… the 5th is clearly redundant and the last one is not very informative, is it ? - In the experiments, rather than describing the Table 1, I would have liked to read the rationale of the choices you made. Why comparing Student’s and Pareto ? Why choosing those parameters and not bigger or smaller ones ? I mean, given that you already give the details in the table, the text could be used to justify those choices. - Experiments again: is it clear that you recover empirically the T^{1/(1+\epsilon)} rate that you obtained theoretically ? That would have deserved a comment.

Reviewer 3



The paper studies linear stochastic bandit problems with heavy-tailed noise. For a noise distribution with finite moments of order 1+epsilon, authors show algorithms with regret bounds that scales as O(T^{1/(1+epsilon)} polylog(T)). One algorithm is based on median of means and the other is based on truncation of payoffs. A matching lower bound of order \Omega(T^{1/(1+epsilon)}) is also shown. In particular, these regret bounds show the optimal scaling of \sqrt{T} for the case of finite variance. Medina and Yang (2016) studied the same problem, but only showed regret bounds that scale as T^{3/4} for the case of epsilon=1 (finite variance). For epsilon=1, regret of MENU (median of means) is O(d^{3/2} \sqrt{T}) while regret of TOFU (truncation) is O(d\sqrt{T}). The TOFU algorithm has a higher computational complexity. The paper is well-written and the theoretical results are very interesting. 1) In Theorem 2, a condition on the noise is assumed while in Theorem 3, a condition on the payoff is assumed. Why the difference? 2) The first algorithm (based on median of means technique) requires playing the same action multiple times. This needs to be stated explicitly in the introduction. Minor comments: * Line 167: Typo: D_t \in R^d * Lines 210-216: Explain more. How do these modifications influence the corresponding regret bounds? * Line 227: Add more details. * Line 249: Typo: MENU -> TOFU * Notation u_{i,t} in Line 9 of Algorithm 2 is not defined.